# Prevalence and incidence of tuberculosis infection among healthcare workers in chest diseases hospitals, Bangladesh: Putting infection control into context

Md. Saiful Islam[1,2¤]*, Emily S. Gurley[1,3], Sayera Banu[1], Kamal Hossain[1], James D. Heffelfinger[4], Kamal Ibne Amin Chowdhury[1], Shahriar Ahmed[1], Sadia Afreen[1], Mohammad Tauhidul Islam[1], Syed Mohammad Mazidur Rahman[1], Arfatur Rahman[1], Michele L. Pearson[4], Shua J. Chai[4]

1 icddr,b, Dhaka, Bangladesh, 2 School of Population Health, University of New South Wales, Sydney, Australia, 3 Johns Hopkins Bloomberg School of Public Health, Baltimore, Maryland, United States of America, 4 Centers for Disease Control and Prevention (CDC), Atlanta, Georgia, United States of America

¤ Current address: School of Population Health, University of New South Wales, Sydney, Australia
* saifsociologist@gmail.com

**Data Availability Statement:** According to icddr,b data sharing policy, data will not be available in public repositories. One copy of the complete

## Abstract

### Background

Healthcare workers (HCWs) are at increased risk of tuberculosis infection (TBI). We estimated the prevalence and incidence of TBI and risk factors among HCWs in Bangladeshi hospitals to target TB infection prevention and control (IPC) interventions.

### Methods

During 2013–2016, we conducted a longitudinal study among HCWs in four chest disease hospitals. At baseline, we administered a questionnaire on sociodemographic and occupational factors for TB, tuberculin skin tests (TST) in all hospitals, and QuantiFERON®-TB Gold in-Tube (QFT-GIT) tests in one hospital. We assessed factors associated with baseline TST positivity (induration ≥10mm), TST conversion (induration increase ≥10mm from baseline), baseline QFT-GIT positivity (interferon-gamma ≥0.35 IU/mL), and QFT-GIT conversion (interferon-gamma <0.35 IU/mL to ≥0.35 IU/mL). We included factors with a biologically plausible relationship with TBI identified in prior studies or having an association (p = <0.20) in the bivariate analyses with TST positivity or QFT-GIT positivity in multivariable generalized linear models. The Kaplan-Meier was used to estimate the cumulative TBI incidence rate per 100 person-years.

### Results

Of the 758 HCWs invited, 732 (97%) consented to participate and 731 completed the one-step TST, 40% had a positive TST result, and 48% had a positive QFT-GIT result. In multivariable models, HCWs years of service 11–20 years had 2.1 (95% CI: 1.5–3.0) times higher odds of being TST-positive and 1.6 (95% CI 1.1–2.5) times higher odds of QFT-GIT-

dataset (anonymized and decoded) and metadata will be shared with the icddr,b repository team after completion of the study. Data access will be subject to the icddr,b data policy (http://www.icddrb.org/policies) upon approval from institutional review board. Interested parties may contact Ms. Armana Ahmed (aahmed@icddrb.org) with further inquiries related to data access. However, a minimal dataset will be made available upon request to the corresponding author at the time of the publication of this article.

**Funding:** This research was funded by the United States Centers for Disease Control and Prevention (CDC), through the cooperative agreement grant number 5U01GH1207. Md Saiful Islam and Emily S Gurley has received the funding. The funders had no role in study design, data collection and analysis, decision to publish, or preparation of the manuscript. The findings and conclusions in this report are those of the authors and do not necessarily represent the official position of CDC.

**Competing interests:** The authors have declared that no competing interests exist.

positivity at baseline compared with those working ≤10 years. HCWs working 11–20 years in pulmonary TB ward had 2.0 (95% CI: 1.4–2.9) times higher odds of TST positivity, and those >20 years had 2.5 (95% CI: 1.3–4.9) times higher odds of QFT-GIT-positivity at baseline compared with those working <10 years. TBI incidence was 4.8/100 person-years by TST and 4.2/100 person-years by QFT-GIT. Females had 8.5 (95% CI: 1.5–49.5) times higher odds of TST conversion than males.

## Conclusions

Prevalent TST and QFT-GIT positivity was associated with an increased number of years working as a healthcare worker and in pulmonary TB wards. The incidence of TBI among HCWs suggests ongoing TB exposure in these facilities and an urgent need for improved TB IPC in chest disease hospitals in Bangladesh.

## Introduction

Tuberculosis (TB) remains an international global health concern, with 10 million people estimated to have become ill with TB disease in 2020, of whom 89% were adults [1]. In Bangladesh, 267,143 new and retreatment TB cases were notified in 2018 [2]. Among notified cases, 81% were pulmonary TB (PTB) infections [2]. The estimated overall incidence rate of TB in Bangladesh was 221 per 100,000 population in 2018, putting Bangladesh in the top ten highest TB burden countries in the globe [2].

Healthcare workers (HCWs) are at risk for occupational exposure to TB [3–6]. In 2019, a systematic review estimated that 46% (95% CI 38–54%) of HCWs in low- and middle-income countries had TB infection (TBI), which is nearly double the global prevalence of TBI [7, 8]. Apriani et al. (2019) also estimated that the yearly incidence of TBI among HCWs by tuberculin skin test (TST) ranged from 1–38% (mean, 17%) and by interferon-gamma release assays (IGRAs), from 10–30% (mean, 18%) [8]. Although the Bangladesh National Tuberculosis Control Program (NTP) recommends treating most TB cases as outpatients using Directly Observed Treatment, Short-course (DOTS), NTP recommends that persons with multi-drug resistant (MDR) TB and complicated TB, such as those with hemoptysis, pleural effusion, cachexia, TB drug intolerance, or other co-morbid conditions, be treated as inpatients at chest disease hospitals, increasing the likelihood of HCWs being exposed to drug-susceptible and drug-resistant TB [9, 10]. Hospitals in Bangladesh often lack basic infection prevention and control (IPC) measures, increasing HCWs' risk for exposure from TB patients and fellow HCWs with undiagnosed PTB [11, 12].

Currently, a gold standard test for the diagnosis of TBI is absent [13]. TBI is diagnosed by measuring the stimulation (*in vivo* or *in vitro*) by *Mycobacterium tuberculosis* complex antigens [14]. TST is performed *in vivo* and has been widely used to estimate TBI prevalence among HCWs in low- and middle-income countries with high TB incidence [15, 16]. However, TST has limitations: it may over-estimate the prevalence of TBI among persons with a history of Bacille Calmette-Guérin (BCG) vaccination or those with exposure to non-tuberculous mycobacteria (NTM) and has low sensitivity in persons with weakened immune systems [17]. IGRAs are performed *in vitro* and have a number of advantages over TST, including no requirement of return visits and being less affected by exposures to BCG vaccination, or by NTM infection [18–20]. However, a major concern about the use of IGRAs is that routine serial testing may lead to false positive conversions (six to nine times more frequent with

IGRAs than with TST) [21]. The WHO recommends either TST or IGRA for TBI detection depending on the country context [22]. Since both TST and IGRA have some advantages and disadvantages in detecting TBI, scientists occasionally use both tests in the same study to screen HCWs for TBI [13, 23].

In Bangladesh, there are limited data on the prevalence and incidence of TBI among HCWs. Determining TBI incidence among HCWs in Bangladesh may help identify high-risk groups or work locations and monitor the effectiveness of IPC interventions to reduce TB transmission risk in hospitals. Bangladesh's NTP guidelines recommend periodic TBI screening of HCWs to document healthcare-associated TB infection and initiate preventive TB treatment [24, 25]. In collaboration with NTP, we sought to estimate the prevalence and incidence of TBI using both IGRA and TST and associated risk factors among HCWs in Bangladeshi chest disease hospitals and identify sub-populations of HCWs at the highest risk of TBI who can be targeted by TB IPC interventions.

## Methods

### Study design and sites

From March 2013 through January 2016, we conducted a prospective longitudinal study in Bangladesh's four largest chest disease hospitals. The study hospitals were located in the Rajshahi (Hospital A), Khulna (Hospital B), Chittagong (Hospital C), and Dhaka (Hospital D) divisions in Bangladesh. Annually, Hospital A admits an average of 1,283, Hospital B 344, Hospital C 289, and Hospital D 3,995 PTB patients as inpatients. These hospitals serve most TB and MDR-TB patients with respiratory complications or co-morbid conditions in the country [9, 26, 27].

### Study definitions, sample size, recruitment, and assessment

A HCW was defined as any person ≥18 years of age who received payment for working in one of the study hospitals. All HCWs at the study hospitals were invited to participate in the study, including ancillary workers (e.g., cleaners, male attendants, and female attendants), clinical staff (e.g., interns, pharmacists, nurses, and doctors), and administrative staff (e.g., clerks and stock/supply chain personnel).

We organized a training session in each of the participating facilities to inform the hospital director, doctors, nurses, and ancillary workers about the study objectives, planned activities, and future use of the data. The field team then collected the hospitals' rosters, listed the names of HCWs by departments and wards, and invited the HCWs to participate during visits to the wards. We recruited HCWs at the hospitals using trained field staff and obtained written informed consent from all enrolled HCWs. We administered a structured questionnaire to collect socio-demographic, professional, and household-level information; social history; and history of past and ongoing TB treatment, including self-reported childhood BCG vaccination.

To assess TBI among HCWs, we used TST alone in three study hospitals (Hospitals A, B, and C) and TST and QFT-GIT in the fourth hospital (Hospital D), which is in the capital city of Dhaka, as QFT-GIT requires laboratory capabilities that are not widely available outside Dhaka. In HCWs who received both tests, the TST was administered immediately after the blood collection for QFT-GIT. To generate an accurate baseline for HCWs, we performed a two-step TST using 0.1ml (2TU) of RT23 purified protein derivative (PPD) under World Health Organization (WHO) recommendations for TST surveys in countries with a high TB burden [28]. The rationale for using two-step TST is that if a person is tested with a single dose of PPD many years after initial infection, the skin test reaction may be negative [29]. A second dose of PPD stimulates a person's ability to react to tuberculin antigen [29]. Trained medical

technicians placed TSTs using the Mantoux method and measured induration size using a caliper at 48–72 hours after PPD placement [30]. For a person whose induration size was < 10mm in the first-TST, we requested that person for a second test after 14 days from the first test [31]. We took their reading following the same procedure used in the first step. If the size was > 10 mm, the person was considered infected [29, 31]. Five HCWs with a history of a prior allergic reaction to PPD were not given a TST at baseline and were excluded from the analyses. For the QFT-GIT test, trained medical technicians collected 3 mL of blood by venipuncture [32]. The TB laboratory team at the International Centre for Diarrhoeal Disease Research, Bangladesh (icddr,b) performed the QFT-GIT assay following the manufacturer's instructions [33, 34]. To determine TBI incidence, we repeated TST testing an average of 24 months (standard deviation 6.1 months) after enrollment for participants with negative baseline TST, and repeated QFT-GIT testing an average of 21 months (standard deviation 3.7 months) after enrollment for participants who had negative baseline QFT-GIT. The field team visited each of the hospital wards several times with the list of HCWs negative at baseline and approached those present to participate in the incidence study.

## Data entry and analysis

Two research team members separately entered collected data into Microsoft Excel and resolved discrepant results by checking the survey questionnaire hardcopies. We defined a QFT-GIT test as positive if the interferon-gamma (IFN-γ) concentration of TB antigen minus Nil was ≥0.35 IU/mL and negative if <0.35 IU/mL, according to the manufacturer's instructions. Since the national HIV prevalence in Bangladesh is <1%, we assumed all study participants were HIV-negative and defined a positive TST as induration ≥10 mm in diameter, consistent with ATS and CDC guidelines for persons working in hospitals and other health care facilities who are not HIV-infected or otherwise immunocompromised [35, 36]. At baseline, a QFT-GIT conversion was defined as an initial IFN-γ level <0.35 IU/mL and a follow-up IFN-γ level ≥0.35 IU/ml, without considering the magnitude of the change of the IFN-γ response [37]. QFT enzyme-linked immunosorbent assay (ELISA) cannot identify absolute values of IFN-γ levels >10 IU/ml, and therefore results >10 IU/ml were recorded as 10 IU/ml. A TST conversion was defined as a ≥10 mm increase in induration diameter on follow-up testing compared with the baseline TST [37]. Participants whose TST results were not read between 48–72 hours after TST placement were excluded from the analyses.

We dichotomized TST and QFT-GIT results as positive or negative. Descriptive statistics were used to summarize the distribution of socio-demographic and clinical features and variables measuring exposure to TB. Odds ratios (ORs) with 95% confidence intervals (CIs) were estimated to determine factors associated with prevalent (one-step and two-step) and incident TST and QFT-GIT positive results in bivariate analyses. Factors with a biologically plausible relationship with TBI (i.e., BCG), confounders identified in prior studies (i.e., age), and biologically plausible associated factors with p<0.20 in the bivariate models with TST positivity or QFT-GIT positivity were included in the multivariable generalized linear models [38]. When the adjusted estimate differed from the unadjusted estimate by approximately 10% or more, we considered that factor as a potential confounder in the final models [39]. We assessed the variance inflation factor (VIF) that measures how the variance of an independent variable is influenced or inflated due to its correlation or interaction with other independent variables in the models, and risk factors with VIF of more than 5 were examined for collinearity and dropped from the models [40]. Risk of TBI is multifactorial, and as a single model is no longer recommended to measure the effects of multiple exposures [41, 42], we constructed new models for each set of variables to enhance the validity of our estimates of effect. We used the

Kaplan-Meier method to estimate cumulative TST conversion rate and cumulative QFT-GIT conversion rate with the outcome variable of time (in months) between a baseline negative test result and a follow-up positive result.

For HCWs who received both TST and QFT-GIT at baseline, we examined the agreement between QFT-GIT and TST results using Cohen's kappa coefficient. We considered k≤ 0 as no agreement, k between 0.01–0.20 as slight agreement, k between as 0.21–0.40 as fair agreement, k between 0.41–0.60 as moderate agreement, and k between 0.61–0.80 as substantial agreement, and k>0.80 as almost perfect agreement [43].

### Ethics statement

The study protocol was reviewed and approved by icddr,b's institutional review board. The U. S. CDC reviewed the protocol and relied on icddr,b's institutional review board approval.

## Results

A total of 1,016 HCWs were listed on rosters of the four participating hospitals, and 758 (75%) were present when the study team visited the hospitals. Of the 758 HCWs present, 732 (97%) consented to and enrolled in the study; two-thirds (498/732) were from Hospital D, the largest chest disease facility in Bangladesh. At Hospital D, 497 (90%) HCWs received a one-step TST, 498 (100%) received QFT-GIT, and 252 (51%) received two-step TST (Fig 1).

The median age of all HCWs at the four hospitals was 42 years (interquartile range [IQR] 14 years); the median age of HCWs who received one-step TST was 43 years (IQR 14 years), the median age of HCWs who received QFT-GIT was 41 years (IQR 14 years) and the median age of the HCWs who received both tests was 41 years (IQR 14 years). About half of the participants were female, and approximately two-third had less than a college education (Table 1). Overall, participating HCWs spent a median of two hours (IQR: 4 hours) per day in PTB wards. No HCWs reported experiencing symptoms of active TB or receipt of TB treatment during the entire study period. However, 3% (23/732) of the HCWs reported being diagnosed with active TB prior to the study period.

### Prevalence of TST positivity

Of 731 HCWs who received a one-step TST at baseline, 296 (40%) had a positive result, ranging from 23% to 46% per site. The median induration of the one-step TST was 8mm (IQR 12). Of the 435 HCWs who were negative at one-step TST, 392 (90%) received two-step TST of whom 21% (81/392) were positive (S1 Table in S1 File). The median induration of the HCWs who were tested positive at the one-step TST was 13mm (IQR10mm-15mm) and those who were tested negative at the first-TST was 3mm (IQR 0mm-7mm). The median induration of the participants who became TST positive in the second-TST was 12mm (IQR11-13). The one-step TST prevalence among HCWs working in Hospital C was 46% (31/68), significantly higher (adjusted odds ratio [aOR] = 3.58, 95% CI: 1.61–7.96) than the other hospitals (Table 2).

In bivariate analyses, study sites, years worked on pulmonary TB wards, longer duration of stay in pulmonary TB wards per day, and longer duration of service as a HCW were significantly associated with two-step TST positivity at baseline. Age of starting work, sex, previous history of BCG vaccination, and ever having had a household member with PTB were not associated with TST positivity (Table 2). In the multivariable models, the HCWs at hospital C (aOR = 3.58, 95% CI: 1.61–7.96), hospital A (aOR = 3.08, 95% CI: 1.37–6.90), and hospital D (aOR = 3.23, 95% CI: 1.70–6.12) had higher odds of positive TST results compared with HCWs at Hospital B (Table 2). Also, HCWs with a history of working 11–20 years and >20

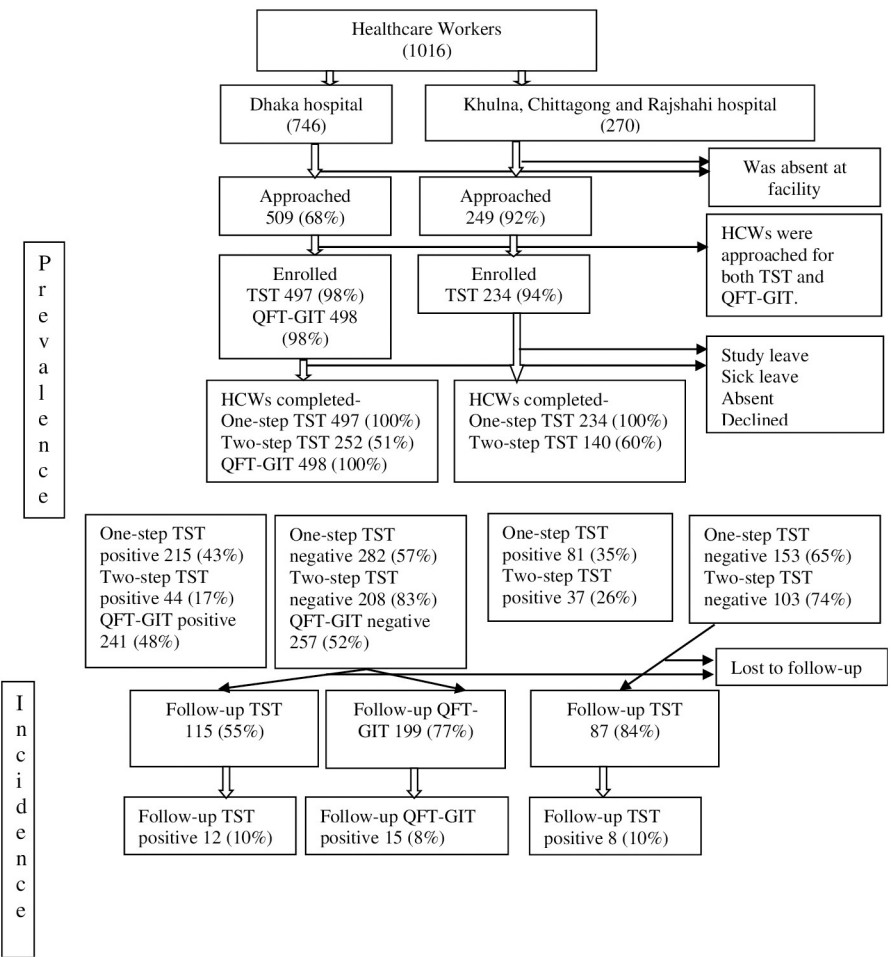

**Fig 1. Study flow chart.**

years on pulmonary TB patients wards had significantly higher odds of TST positivity (aOR = 2.01, 95% CI: 1.39–2.93 and aOR = 2.09, 95% CI: 1.26–3.45) compared with HCWs working less than 10 years on the same wards (Table 2). Moreover, HCWs spending more than two hours a day in pulmonary TB wards had significantly higher odds of TST positivity (aOR = 1.91, 95% CI: 1.37–2.68). Duration of service as a HCW 11–20 years (aOR 2.09, 95% CI: 1.48–2.97) and >20 years (aOR 2.05, 95% CI: 1.28–3.28) were also associated with higher odds of TST positivity compared with those ≤10 years(Table 2).

## TST conversion rate

Among the 311 HCWs who had a negative TST at baseline (negative by one-step and two-step TST), 202 (65%) received a follow-up TST (Fig 1), and 109 (35%) were lost to follow up because they retired from work, were transferred to other facilities, or refused to participate in the study. S2 Table in S1 File showed that HCWS who were lost to follow up for TST statistically significantly varied by location of hospitals, education, and age of starting work. Twenty (10%) HCWs had TST conversions for a cumulative TST conversion rate of 4.8 per 100 person-years (95% CI: 3.08–7.42) (Table 3). Female HCWs spent on average of 1.42 hours (95% CI: 1.12–1.73) more per day on a PTB ward compared with male HCWs in the unadjusted

**Table 1. Baseline characteristics of healthcare workers at four chest diseases hospitals in Bangladesh, 2013.**

| Characteristic | TST participants (N = 689) % (n) | QFT-GIT participants (N = 498) % (n) |
|---|---|---|
| **Hospitals** | | |
| Hospital A | 11 (78) | |
| Hospital B | 12 (80) | |
| Hospital C | 9 (63) | |
| Hospital D | 68 (467) | 100 (498) |
| **Sex** | | |
| Male | 44 (306) | 47 (236) |
| Female | 56 (382) | 53 (262) |
| **History of BCG vaccination** | | |
| Yes | 80 (548) | 81(404) |
| No | 18 (125) | 17 (87) |
| Don't know | 2 (15) | 1 (7) |
| **Highest education completed** | | |
| None to Primary | 9 (64) | 9 (44) |
| Secondary | 22 (149) | 23 (114) |
| Higher Secondary | 43 (297) | 37 (183) |
| Bachelor and above | 26 (178) | 32 (157) |
| **Profession** | | |
| Doctors[i] and pharmacists | 9 (65) | 12 (58) |
| Nurse | 42 (290) | 36 (180) |
| Admin workers | 18 (123) | 21 (103) |
| Laboratory Staff | 6 (39) | 7 (35) |
| Ancillary workers | 25 (171) | 24 (122) |
| **Age of starting work, years[ii]** | | |
| Median, IQR | 24 (21–27) | 24 (21–27) |
| <20 | 12 (85) | 13 (67) |
| 21–25 | 46 (314) | 45 (222) |
| 26–30 | 32 (217) | 31 (154) |
| >30 | 10 (71) | 11 (54) |
| **Years worked in pulmonary TB ward[iii]** | | |
| ≤ 10 | 57 (323) | 59 (245) |
| 11–20 | 29 (168) | 30 (126) |
| >20 | 14 (82) | 11 (47) |

model. In the multivariable model, only female sex was associated with higher odds of developing new TBI (aOR = 8.50; 95% CI: 1.46–49.48) after adjusting for age of starting work, history of BCG vaccination, history of living with someone diagnosed with pulmonary TB, and profession (Table 3).

## Prevalence of QFT-GIT positivity

Of 498 HCWs tested by QFT-GIT at baseline, 241 (48%) had a positive result. The highest prevalence was among ancillary workers (59%) (Table 4). In bivariate analyses, occupational group, level of education, years working in a PTB ward, years of service as a HCWs, and any use of a facemask or N95 respirator during contacts with TB patients were significantly associated with QFT-GIT positivity at baseline (Table 4). In the multivariable model, a HCW with a history of working >20 years in pulmonary TB patients ward had significantly higher odds of

**Table 2. Factors associated with one-step TST positivity among HCWs in four chest diseases hospitals, Bangladesh, 2013 (N = 731).**

| | TST positive % (n/N) | TST negative % (n/N) | OR (95% CI) | aOR (95% CI) |
|---|---|---|---|---|
| **Location of hospital[i]** | | | | |
| Chittagong | 46 (31/68) | 54 (37/68) | **2.82 (1.40–3.96)** | **3.58 (1.61–7.96)** |
| Rajshahi | 37 (31/83) | 63 (52/83) | **2.01 (1.02–3.96)** | **3.08 (1.37–6.90)** |
| Dhaka | 43 (215/497) | 57 (282/497) | **2.57 (1.49–4.42)** | **3.23 (1.70–6.12)** |
| Khulna | 23 (19/83) | 77 (64/83) | Reference | Reference |
| **Sex** | | | | |
| Male | 38 (125/329) | 62 (204/329) | 0.83 (0.61–1.11) | |
| Female | 43 (171/402) | 57 (231/402) | Reference | |
| **History of BCG vaccination** | | | | |
| Yes | 41 (242/585) | 59 (343/585) | 1.21 (0.81–1.78) | |
| No | 37 (48/130) | 63 (82/130) | Reference | |
| Don't know | 38 (6/16) | 62 (10/16) | 1.03 (0.35–3.00) | |
| **Profession** | | | | |
| Doctors including interns and residents and pharmacists* | 32 (26/82) | 60 (56/82) | 0.89 (0.49–1.59) | |
| Nurse | 43 (129/299) | 57 (170/299) | 1.45 (0.95–2.22) | |
| Admin Officer | 34 (45/131) | 66 (86/131) | Reference | |
| Laboratory Staff | 40 (17/43) | 60 (26/43) | 1.25 (0.61–2.54) | |
| Ancillary workers | 45 (79/176) | 55 (97/176) | 1.56 (0.98–2.48) | |
| **Highest education completed** | | | | |
| None to Primary | 40 (27/67) | 60 (40/67) | 1.10 (0.63–1.94) | |
| Secondary | 41 (63/154) | 59 (91/154) | 1.13 (0.74–1.74) | |
| Higher Secondary | 42 (129/307) | 58 (178/307) | 1.19 (0.82–1.70) | |
| Bachelor and above | 38 (77/203) | 62 (126/203) | Reference | |
| **Years worked on pulmonary TB patient wards[ii]** | | | | |
| ≤10 | 34 (117/347) | 66 (230/347) | Reference | **Reference** |
| 11–20 | 51 (89/175) | 49 (86/175) | **2.03 (1.40–2.95)** | **2.01 (1.39–2.93)** |
| >20 | 51 (43/85) | 49 (42/85) | **2.01 (1.25–3.25)** | **2.09 (1.26–3.45)** |
| **Hours worked on pulmonary TB wards per day[iii]** | | | | |
| ≤2 | 34 (94/273) | 66 (179/273) | Reference | Reference |
| >2 | 50 (155/308) | 50 (153/308) | **1.93 (1.38–2.70)** | **1.91 (1.37–2.68)** |
| **Years of service as a healthcare worker[iv]** | | | | |
| ≤ 10 | 35 (139/399) | 65 (260/399) | Reference | Reference |
| 11–20 | 52 (108/207) | 48 (99/207) | **2.04 (1.45–2.87)** | **2.09 (1.48–2.97)** |
| >20 | 49 (49/101) | 51 (52/101) | **1.76 (1.13–2.74)** | **2.05 (1.28–3.28)** |
| **Age of starting work, years** | | | | |
| <20 | 44 (40/90) | 56 (50/90) | Reference | |
| 21–25 | 39 (128/327) | 61 (199/327) | 0.80 (0.50–1.29) | |
| 26–30 | 40 (96/238) | 60 (142/238) | 0.85 (0.52–1.38) | |
| >30 | 43 (32/75) | 57 (43/75) | 0.93 (0.50–1.73) | |
| **Lived with someone diagnosed with pulmonary TB** | | | | |
| Yes | 41 (34/83) | 59 (47/83) | 1.03 (0.64–1.64) | |
| No | 40 (258/640) | 60 (382/640) | Reference | |
| Don't know | 57 (4/7) | 43 (3/7) | 1.97 (0.44–8.89) | |
| **Ever used a face mask or N 95 respirator[v]** | | | | |
| Yes | 45 (147/329) | 55 (182/329) | **1.37 (1.02–1.85)** | **1.68 (1.16–2.43)** |

*(Continued)*

**Table 2.** (Continued)

| | TST positive % (n/N) | TST negative % (n/N) | OR (95% CI) | aOR (95% CI) |
|---|---|---|---|---|
| No | 37 (149/402) | 63 (253/402) | Reference | |

OR = Odds Ratio, CI = Confidence interval, aOR = Adjusted Odds Ratio

[i] Adjusted for the years of work in the pulmonary TB ward, starting age at work in years, history of BCG vaccine, ever lived with someone diagnosed with pulmonary TB, and ever used a mask or respirator in the ward.

[ii] Adjusted for lived with someone diagnosed with pulmonary TB and age in years.

Adjusted for the history of BCG vaccine, lived with someone diagnosed with pulmonary TB, and starting age at work in years.

[iii] Adjusted for the history of BCG vaccine, lived with someone diagnosed with pulmonary TB and starting age at work in years.

[iv] Adjusted for the location of hospitals, starting age at work in years, history of BCG vaccine, and ever lived with someone diagnosed with pulmonary TB.

[v] Adjusted for the location of hospitals, starting age at work in years, history of BCG vaccine, ever lived with someone diagnosed with pulmonary TB and the years of work in the pulmonary TB ward

QFT-GIT positivity (aOR = 2.48, 95% CI: 1.25–4.90) compared with HCWs working ≤10 years. HCWs with 11–20 (aOR 1.60, 95% CI: 1.05–2.45) and >20 (aOR 3.13, 95% CI: 1.62–6.04) years of service had significantly higher odds of QFT-GIT positivity compared to HCWs with ≤10 years. Similarly, any use of a facemask or N95 respirator during contacts with TB patients was associated with higher odds (aOR 2.01, 95% CI: 1.39–2.88) of QFT-GIT positivity (Table 4). S1 Table in S1 File showed that none of the factors were statistically significantly associated with two-step TST.

### QFT-GIT conversion rate

Among the 257 HCWs who had negative QFT-GIT results at baseline, 199 (77%) had a follow-up QFT-GIT test. And 58 (23%) were lost to follow up because they retired from work, were transferred to other facilities, or refused to participate in the study. S2 Table in S1 File showed that there was no statistically significant difference in HCWs' characteristics who participated in the follow-up QFT-GIT and who were lost to follow up. Fifteen (8%) had QFT-GIT conversions for a cumulative QFT-GIT conversion rate of 4.2 per 100 person-years. No factors were significantly associated with QFT-GIT conversion in the bivariate or the multivariable model (Table 5).

### TST and QFT-GIT agreement

497 HCWs were tested with both TST and QFT-GIT. Using QFT-GIT as a gold standard, the agreement between TST and QFT-GIT was poor (Cohen's kappa 0.33, 95% CI 0.25–0.42) when comparing baseline results. S3 Table in S1 File shows the results of the analysis of concordance and discordance between the one-step TST and the QFT-GIT test. QFT-GIT only positive results were significantly associated with the number of years working as HCWs after controlling for sex, age of starting work, history of BCG vaccine, and history of having a household member with pulmonary TB. In contrast, TST-only positive results had no association with the exposures in the model. Having both TST and QFT positive results were significantly associated with years working as HCWs and any use of a facemask or N95 respirator during contacts with TB patients after controlling for sex, age of starting work, history of BCG vaccine, and history of having a household member with pulmonary TB.

### Discussion

During a longitudinal study over nearly two years, we identified a high baseline prevalence of TBI, ranging from 40% (by one-step TST) to 48% (by QFT-GIT). These measures of

**Table 3. Factors associated with incident TST positivity among healthcare workers in four chest diseases hospitals, Bangladesh, 2013–2016.**

| | TST positive % (n/N) | TST negative % (n/N) | OR (95% CI) | aOR(95% CI) |
|---|---|---|---|---|
| **Location of hospital** | | | | |
| Chittagong | 0 (0/15) | 100 (15/15) | undefined | |
| Rajshahi | 15 (5/33) | 85 (28/33) | 2.14 (0.47–9.74) | |
| Dhaka | 10 (12/115) | 90 (103/115) | 1.40 (0.37–5.24) | |
| Khulna | 8 (03/39) | 92 (36/39) | Reference | |
| **Sex[i]** | | | | |
| Male | 04 (3/85) | 96 (82/85) | Reference | Reference |
| Female | 15 (17/117) | 85 (100/117) | **4.65 (1.32–16.41)** | **8.50 (1.46–49.48)** |
| **History of BCG vaccination** | | | | |
| Yes | 09 (15/161) | 91 (146/161) | Reference | |
| No | 11 (4/36) | 89 (32/36) | 1.22 (0.38–3.90) | |
| Don't know | 20 (1/5) | 80 (4/5) | 2.43 (0.26–23.20) | |
| **Profession** | | | | |
| Doctorsincluding interns and residents and pharmacist | 14 (2/14) | 86 (12/14) | 1.46 (0.24–9.00) | |
| Nurse | 15 (14/94) | 85 (80/94) | 1.53 (0.47–4.98) | |
| Admin Officer | 10 (4/39) | 90 (35/39) | Reference | |
| Laboratory Staff | 0.0 (0/11) | 100 (11/11) | Undefined | |
| Ancillary workers | 00 (0/44) | 100 (44/44) | Undefined | |
| **Highest education completed** | | | | |
| None to Primary | 06 (1/16) | 94 (15/16) | 0.84 (0.08–8.77) | |
| Secondary | 07 (3/42) | 93 (39/42) | 0.97 (0.18–5.13) | |
| Higher Secondary | 13 (13/103) | 87 (90/103) | 1.83 (0.49–6.79) | |
| Bachelor and above | 07 (3/41) | 93 (38/41) | Reference | |
| **Year worked on pulmonary TB wards** | | | | |
| 0–10 | 8 (8/105) | 92 (97/105) | Reference | |
| 11–20 | 16 (6/38) | 84 (32/38) | 2.27 (0.73–7.05) | |
| >20 | 10 (2/20) | 90 (18/20) | 1.35 (0.26–6.87) | |
| **Hours worked on pulmonary TB patients wards per day** | | | | |
| ≤2 | 10 (9/87) | 90 (78/87) | Reference | |
| >2 | 9 (7/76) | 91 (69/76) | 0.88 (0.31–2.49) | |
| **Years of service as healthcare worker** | | | | |
| ≤ 10 | 9 (11/130) | 91 (119/130) | Reference | |
| 11–20 | 13 (6/45) | 87 (39/45) | 1.66 (0.58–4.80) | |
| >20 | 11 (3/27) | 89 (24/27) | 1.35 (0.35–5.22) | |
| **Age of starting work in years [vi]** | | | | |
| <20 | 19 (4/21) | 81 (17/21) | Reference | |
| 20–25 | 11 (12/105) | 89 (93/105) | 0.55 (0.16–1.90) | |
| 26–30 | 6 (4/63) | 94 (59/63) | 0.29 (0.07–1.27) | |
| >30 | 0 (0/13 | 100 (13/13) | ~ | |
| **Lived with someone diagnosed with pulmonary TB** | | | | |
| Yes | 14 (3/22) | 86 (19/22) | 1.50 (0.40–5.58) | |
| No | 10 (17/178) | 90 (161/178) | Reference | |
| Don't know | 0 (0/2) | 100 (2/2) | Undefined | |
| **Ever used a face mask or N95 respirator** | | | | |
| Yes | 7 (6/81) | 93 (75/81) | Reference | |

(*Continued*)

**Table 3.** (Continued)

|  | TST positive % (n/N) | TST negative % (n/N) | OR (95% CI) | aOR(95% CI) |
|---|---|---|---|---|
| No | 12 (14/121) | (107/121) | 1.64 (0.60–4.45) |  |

OR = Odds Ratio, CI = Confidence interval, aOR = Adjusted Odds Ratio

[i] Adjusted for history of BCG vaccine, profession, and age of starting to work in years.

prevalence were nearly double the prevalence estimated for general populations reported from studies around the world [7]. The TST prevalence measured by QFT-GIT in this study population was higher than the 42% previously reported among HCWs in public tertiary care general hospitals in Bangladesh that admitted TB patients occasionally for a shorter time [44]. Conversely, this TBI prevalence was similar to QFT-GIT results from a pooled prevalence of TBI of 47% (95% CI 34–60%) estimated in a systematic review and meta-analysis in TB high burden countries [45]. The factors associated with TBI prevalence suggest occupational exposures were important, irrespective of which test was used, and the HCWs at the chest diseases hospitals are at increased risk compared to general hospitals.

The incidence of TBI differed by mode of testing, ranging from 4.2 (by QFT-GIT) to 4.8 (by TST) in all hospitals per 100 person-years. These rates were lower than those found in a study among medical residents and nursing students in Pune city, India, where the annual rate of infection was 28.7 (95% CI, 20.6–38.9) per 100 person-years using TST with a similar ≥10mm induration cutoff and 17.4 (95% CI, 11.5–25.4) per 100 person-years by QFT-GIT [46]. In our study, none of the incident cases detected by TST or QFT-GIT was among medical residents or interns—in fact, the average number of years worked as a HCW was 16.96, much longer than among medical residents or nursing students. The difference in incidence was expected as the Pune study has a baseline TBI prevalence of 30% which is much lower than our baseline prevalence of 48% (by QFT-GIT) among HCWs in our study which suggests a much larger cohort that would be susceptible to TB infection among the cohort of trainee HCWs in India. The high incidence rate of TBI in Bangladesh, despite the high background prevalence, suggests continuing exposure of these seasoned HCWs to TB in these hospitals. In addition, Bangladesh had an estimated national TB incidence rate of 221 per 100,000 population in 2018, which is higher than the rate of Pune, India (185 per 100,000); as lapses in TB IPC are common and TB patients spend considerable time in the hospital in Bangladesh, HCWs likely have a similarly high risk of encountering TB in their occupational settings [2, 47].

Our study identified occupational factors statistically significantly associated with TBI infection, and the findings are consistent with results reported in other studies [4, 48]. Prevalence of TBI detected using both TST and QFT-GIT were independently associated with greater years of service as HCWs and greater years of service on pulmonary wards after controlling for age of starting work, history of BCG, and a history of living with someone diagnosed with pulmonary TB. These results are supported by prior studies, which similarly showed that the risk of TBI increases with increased length of service [49, 50]. More specifically, TBI prevalence by TST was higher among HCWs who spent more than two hours in pulmonary TB wards per day as well as those who worked 11–20 and >20 years compared with those working ≤10 years, further supporting the hypothesis that longer duration of exposure to pulmonary TB patients increases TB infection risk [51]. The similar finding of increased odds of TBI prevalence by QFT-GIT among HCWs working >20 years on pulmonary TB patient wards is consistent with the findings from the TST results. We found that chest disease hospitals have a high burden of TB patients, and these hospitals have been shown to lack basic

**Table 4. Factors associated with prevalent QFT-GIT positivity among healthcare workers in Hospital D, Bangladesh, 2013–2016.**

| | QFT-GIT positive % (n/N) | QFT-GIT Negative % (n/N) | OR(95% CI) | aOR (95% CI) |
|---|---|---|---|---|
| **Sex** | | | | |
| Male | 46 (109/236) | 54 (127/236) | 0.85 (0.59–1.20) | |
| Female | 50 (132/262) | 50 (132/262) | Reference | |
| **History of BCG vaccination** | | | | |
| Yes | 47 (191/404) | 53 (213/404) | 0.73 (0.45–1.16) | |
| No | 55 (48/87) | 45 (39/87) | Reference | |
| Don't know | 29 (2/7) | 71 (5/7) | 0.33 (0.06–1.77) | |
| **Profession[i]** | | | | |
| Doctorincluding interns and residents and pharmacist | 45 (26/58) | 55(32/58) | 1.01 (0.53–1.92) | 0.85 (0.43–1.68) |
| Nurse | 47 (84/180) | 53 (96/180) | 1.08 (0.67–1.76) | 0.95 (0.56–1.61) |
| Admin Officer | 45 (46/103) | 55 (57/103) | Reference | Reference |
| Laboratory Staff | 37 (13/35) | 63 (22/35) | 0.73 (0.33–1.61) | 0.63 (0.28–1.44) |
| Ancillary workers | 59 (72/122) | 41 (50/122) | **1.78 (1.05–3.03)** | 1.26 (0.71–2.24) |
| **Highest education completed[ii]** | | | | |
| None to Primary | 73 (32/44) | 27 (12/42) | **3.58 (1.72–7.47)** | 2.40 (0.90–6.38) |
| Secondary | 51 (58/114) | 49 (56/114) | 1.39 (0.86–2.26) | 1.11 (0.53–2.36) |
| Higher Secondary | 46 (84/183) | 54 (99/183) | 1.14 (0.74–1.75) | 0.11 (0.64–1.91) |
| Bachelor and above | 43 (67/157) | 57 (90/157) | Reference | Reference |
| **Years worked on pulmonary TB patient wards[iii]** | | | | |
| ≤10 | 46 (113/245) | 54 (132/245) | Reference | Reference |
| 11–20 | 52 (65/126) | 48 (61/126) | 1.24 (0.81–1.91) | 1.23 (0.79–1.91) |
| >20 | 66 (31/47) | 34 (16/47) | **2.26 (1.18–4.35)** | **2.48 (1.25–4.90)** |
| **Hours worked on pulmonary TB patients ward per day** | | | | |
| ≤2 | 49 (68/139) | 51 (71/139) | Reference | |
| >2 | 51 (131/257) | 49 (126/257) | 1.09 (0.72–1.64) | |
| **Years of service as healthcare worker[iv]** | | | | |
| ≤ 10 | 44 (118/271) | 56 (153/271) | Reference | Reference |
| 11–20 | 54 (81/151) | 46 (70/151) | 1.50 (1.01–2.24) | **1.60 (1.05–2.45)** |
| >20 | 69 (37/54) | 31 (17/54) | **2.82 (1.51–5.26)** | **3.13 (1.62–6.04)** |
| **Age of starting work in years [v]** | | | | |
| ≥20 | 54 (36/67) | 46 (31/67) | Reference | Reference |
| 21–25 | 43 (95/222) | 57 (127/222) | 0.64 (0.37–1.11) | 0.76 (0.39–1.47) |
| 26–30 | 48 (74 /154) | 52 (80 /15) | 0.80 (0.45–1.42) | 0.96 (0.49–1.90) |
| >30 | 65 (35/54) | 35 (19/54) | 1.59 (0.76–3.31) | 1.74 (0.76–3.95) |
| **Lived with someone diagnosed with pulmonary TB** | | | | |
| Yes | 53 (31/59) | 47 (28/59) | 1.21 (0.70–2.09) | |
| No | 48 (206/431) | 52 (225/431) | Reference | |
| Don't know | 43 (3/7) | 57 (4/7) | 0.82 (0.18–3.70) | |
| **Ever used a face mask or N 95 respirator[vi]** | | | | |
| Yes | 58 (120/207) | 42 (87/207) | **1.94 (1.35–2.78)** | **2.01 (1.39–2.88)** |
| No | 42 (121/291) | 58 (170/291) | Reference | Reference |

OR = Odds Ratio, CI = Confidence interval, aOR = Adjusted Odds Ratio

[i] Adjusted for history of BCG vaccine, age of starting work in years, lived with someone diagnosed with pulmonary TB, and ever used a face mask or N95 respirators.

[ii] Adjusted for lived with someone diagnosed with pulmonary TB, profession, age of starting work in years and ever used a face mask or N95 respirators.

[iii] Adjusted for history of BCG vaccine, age of starting work in years and lived with someone diagnosed with pulmonary TB.

[iv] Adjusted for history of BCG vaccine, age of starting work in years, profession, and lived with someone diagnosed with pulmonary TB.

[v] Adjusted for highest education completed, lived with someone diagnosed with pulmonary TB, profession, and ever used a face mask or N95 respirators.

[vi] Adjusted for age of starting work in years, and lived with someone diagnosed with pulmonary TB.

**Table 5. Factors associated with incident QFT-GIT positivity among healthcare workers by exposures in hospital D, Bangladesh, 2013–2016.**

| Characteristic | QFT-GIT positive % (n/N) | QFT-GIT negative % (n/N) | OR (95% CI) | aOR (95% CI) |
|---|---|---|---|---|
| **Sex** | | | | |
| Male | 6 (6/94) | 94 (88/94) | Reference | |
| Female | 9 (9/105) | 91 (96/105) | 1.37 (0.47–4.02) | |
| **History of BCG vaccination** | | | | |
| Yes | 5 (9/165) | 95 (156/165) | Reference | Reference* |
| No | 17 (5/30) | 83 (25/30) | **3.47 (1.07–11.19)** | 3.01 (0.91–9.93) |
| Don't know | 25 (1/04) | 75 (3/4) | 5.78 (0.55–61.24) | 5.95 (0.55–64.69) |
| **Profession** | | | | |
| Doctorincluding interns and residents and pharmacists | 19 (3/16) | 81 (13/16) | 5.08 (0.76–33.71) | |
| Nurse | 8 (6/77) | 92 (71/77) | 1.86 (0.36–9.62) | |
| Admin Officer | 4 (2/46) | 96(44/46) | Reference | |
| Laboratory Staff | 6 (1/17) | 94 (16/17) | 1.36 (0.12–16.22) | |
| Ancillary workers | 7 (3/43) | 93 (40/43) | 1.65 (0.26–10.39) | |
| **Highest education completed** | | | | |
| None to Primary | 18 (2/11) | 82 (09/11) | 3.06 (0.49–19.20) | |
| Secondary | 7 (3/46) | 93 (43/46) | 0.96 (0.20–4.52) | |
| Higher Secondary | 7 (6/83) | 93 (77/83) | 1.07 (0.29–3.98) | |
| Bachelor and above | 7 (4/59) | 93 (55/59) | Reference | |
| **Year worked on pulmonary TB wards** | | | | |
| 0–10 | 7 (7/95) | 93 (88/95) | Reference | |
| 11–20 | 14 (7/51) | 86 (44/51) | 2.00 (0.66–6.06) | |
| >20 | 0 (0/15) | 100 (15/15) | Undefined | |
| **Hours worked on pulmonary TB patients ward per day** | | | | |
| ≤2 | 12 (6/51) | 88 (45/51) | Reference | |
| >2 | 8 (8/106) | 92 (98/106) | 0.61 (0.20–1.89) | |
| **Years of service as healthcare workers** | | | | |
| ≤ 10 | 7 (8/117) | 93 (109/117) | Reference | |
| 11–20 | 9 (6/59) | 90 (53/59) | 1.54 (0.51–4.67) | |
| >20 | 0 (0/16) | 100 (16/16) | Undefined | |
| **Age of starting work in years** | | | | |
| ≤20 | 17 (04/24) | 83 (20/24) | Reference | |
| 21–25 | 7 (7/95) | 93 (88/95) | 0.40 (0.11–1.49) | |
| 26–30 | 5 (3/66) | 95 (63/66) | 0.24 (0.05–1.16) | |
| >30 | 7 (1/14) | 93 (13/14) | 0.38 (0.04–3.84) | |
| **Lived with someone diagnosed with pulmonary TB** | | | | |
| Yes | 5 (1/21) | 95 (20/21) | 0.58 (0.07–4.61) | |
| No | 8 (14/175) | 92 (161/175) | Reference | |
| Don't know | 0 (0/3) | 100 (3/3) | Undefined | |
| **Ever used a face mask or N 95 respirator** | | | | |
| Yes | 7 (5/68) | 93 (63/68) | 0.96 (0.31–2.93) | |
| No | 8 (10/131) | 92 (121/131) | Reference | |

OR = Odds Ratio, CI = Confidence interval;

*Adjusted for age of starting work in years, and lived with someone diagnosed with pulmonary TB.

TB IPC measures, including TB infection control committees, isolation rooms, regular supplies of N95 respirators, and routine training for HCWs on TB infection control [12, 52].

Predictably, the prevalence of TBI measured by TST and QFT-GIT was not statistically significantly associated with age of starting work further supporting occupational TB exposures. The one community TB exposure among HCWs that we did measure in this study, history of household contact with a person with TB disease, was also not associated with an increased prevalence of TBI, possibly owing to the small number of HCWs who reported this exposure. Furthermore, additional healthcare-related occupational exposures were not captured in our data. Public sector HCWs often work in private practice settings and clinics, and the time spent in these other jobs are likely to be substantial [53]. We do not know the IPC practices in private clinics; studies exploring private practices and clinics' practices might generate a more complete picture of HCWs' exposures in all workplaces.

While the number of PTB patients under treatment differs by the hospital, the availability of effective tuberculosis IPC measures can reduce transmission and is likely an important contributor to the difference seen in TBI prevalence identified by TST in hospitals. In an earlier evaluation of TB IPC healthcare measures in the participating chest disease hospitals, implementation gaps, including triaging of presumptive TB patients and use of masks by TB patients, varied by hospital [54]. The variations found in TBI prevalence by study sites were consistent with the variation found in IPC practices in those hospitals in a previous study and consistent with the findings from a study conducted in high TB burden countries [52, 55]. A national evaluation of the implementation of TB IPC measures in hospitals and other healthcare facilities could identify common gaps that could be addressed to decrease nosocomial TB transmission.

Our study identified an association between increased prevalence of TBI (by both TST and QFT-GIT) and the use of masks or N95 respirators. The use of masks and N95 respirators in this study might be a marker for a HCW working in an area with greater risk for TB exposure, taking greater risks while wearing only a mask, or wearing N95 respirators improperly, as masks and irregular or improper use of respirators provide no actual protection against TB [44]. Nazneen et al. (2021) found a limited supply of N95 respirators in the hospitals, and the use of N95 respirators varied by category of profession, which might have resulted in irregular use of respirators among HCWs [12]. In chest disease hospitals in Bangladesh, HCWs spend six to eight hours a day on inpatient wards with limited or no respiratory protection [56] while providing care to TB patients, cleaning rooms of TB patients, and accompanying TB patients to radiological studies and DOTS facilities [56, 57]. Similar findings have also been reported in a study in Brazil [58], where authors found that irregular use of N95 respirators did not provide protection, but rather increased the risk of QFT-GIT positivity by more than two and a half times.

Females had a higher TBI incidence even after controlling for profession, history of BCG and living with someone who had TB. A similar difference, albeit nonsignificant, was also observed in TBI incidence detected using QFT-GIT. This finding is consistent with a study conducted among HCWs in public tertiary-care general hospitals in Bangladesh, where females were 1.08 times more likely to be TST positive when compared with males [44]. Our findings are also similar to those reported from a study conducted among HCWs in South Africa where females had higher rates of new TB infections—30 per 100 person-years by TST and 37 per 100 person-years by QFT-GIT, and for males, 5 per 100 person-years by TST and 12 per 100 person-years by QFT-GIT [59]. While the reasons for the elevated incidence among females are unclear, the results suggest higher exposures among females in inpatient wards in both general and chest disease hospitals in Bangladesh.

The discordance identified between TST and QFT-GIT might be attributable to several factors, including the effect of age on reactivity to TST, BCG vaccination status, and exposure to NTM, which are prevalent in Bangladesh [13, 60]. We found a lower prevalence of TST

positivity among HCWs who started working at >30 years of age compared to other ages, while QFT-GIT positivity increased consistently with increasing age; this pattern is similar to results reported by other studies, where the decreased reaction to TST was attributed to the decreased ability of elderly skin to react [13, 61]. The findings in our study support recent findings that suggest that QFT-GIT is more useful in detecting TBI among elderly populations than TST [62]. We could not systematically exclude all TB exposures in the community. However, years working as a HCW in a chest disease hospital, years working on pulmonary TB patient wards, and hours working on PTB patient wards per day remained significantly associated with TBI after controlling for non-workplace factors (history of BCG vaccine, and living with someone who had TB). A second limitation was participation attrition; 35% of HCWs were lost to follow up for the repeat TST and 23% for the repeat QFT-GIT due to job transfer, illness, retirement from work, and unwillingness to participate. These participants lost to follow-up could have developed a new infection, and therefore, our study could have inaccurately estimated the true incidence of TST and QFT-GIT positivity. In addition, owing to the loss of follow-up, the incidence analysis was underpowered to detect true differences between groups, particularly by professional category. Finally, we did not obtain chest radiographs on HCWs positive with TST or QFT-GIT. We only relied on TB symptoms screening and might have missed cases with active TB.

This study estimated the prevalence of, the incidence of, and factors associated with TBI among HCWs in chest disease hospitals in Bangladesh for the first time. The data can be used as a baseline for future interventions. Healthcare workers with TBI represent a substantial reservoir for future cases of TB disease, and therefore the NTP might consider providing TBI preventive treatment. As the study hospitals admit and treat MDR-TB patients regularly, NTP should consider this when deciding whether to offer HCWs preventive treatment and which regimen to use. In addition, there is an urgent need to establish effective TB IPC programs in healthcare facilities to reduce nosocomial TB transmission and protect HCWs and other ward occupants. An assessment of the implementation of Bangladesh's national TB IPC guidelines to identify modifiable administrative and engineering measures, such as improving facility layout and patient flows for preventing airborne pathogen transmission and reducing barriers to personal protective equipment use, can help protect HCWs and prevent TB transmission in chest disease hospitals and other healthcare facilities in Bangladesh. We also do not have rigorous estimates of the prevalence and incidence of TBI among the general population in Bangladesh. Future surveys that target estimating TBI prevalence and incidence among the general population would help estimate the true contribution of occupational exposures to TBI among HCWs.

## Supporting information

**S1 File.**
(DOCX)

## Acknowledgments

We would like to thank all the study participants for their time and respect. icddr,b is grateful to the Governments of Bangladesh, Canada, Sweden, and the UK for providing core/unrestricted support. We also acknowledge Arkray Healthcare Pvt Ltd for their support.

## Author Contributions

**Conceptualization:** Md. Saiful Islam, Emily S. Gurley, James D. Heffelfinger, Shahriar Ahmed, Sadia Afreen, Michele L. Pearson, Shua J. Chai.

**Data curation:** Md. Saiful Islam, Kamal Hossain.

**Formal analysis:** Md. Saiful Islam, Emily S. Gurley, Sayera Banu, Kamal Hossain, James D. Heffelfinger, Kamal Ibne Amin Chowdhury, Shahriar Ahmed, Sadia Afreen, Arfatur Rahman, Michele L. Pearson, Shua J. Chai.

**Funding acquisition:** Md. Saiful Islam, Shua J. Chai.

**Investigation:** Md. Saiful Islam, Kamal Ibne Amin Chowdhury, Shahriar Ahmed, Syed Mohammad Mazidur Rahman.

**Methodology:** Md. Saiful Islam, Emily S. Gurley, Sayera Banu, Sadia Afreen, Syed Mohammad Mazidur Rahman, Arfatur Rahman, Michele L. Pearson, Shua J. Chai.

**Project administration:** Md. Saiful Islam, Mohammad Tauhidul Islam.

**Resources:** Mohammad Tauhidul Islam.

**Supervision:** Sayera Banu, Michele L. Pearson, Shua J. Chai.

**Validation:** Kamal Hossain, Mohammad Tauhidul Islam, Syed Mohammad Mazidur Rahman.

**Writing – original draft:** Md. Saiful Islam.

**Writing – review & editing:** Md. Saiful Islam, Emily S. Gurley, Sayera Banu, Kamal Hossain, James D. Heffelfinger, Kamal Ibne Amin Chowdhury, Shahriar Ahmed, Sadia Afreen, Mohammad Tauhidul Islam, Syed Mohammad Mazidur Rahman, Arfatur Rahman, Michele L. Pearson, Shua J. Chai.

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
