## [Decision Letter · Decision Letter 0]

8 Nov 2022

PONE-D-22-14779Prevalence and incidence of latent tuberculosis infection among healthcare workers in chest diseases hospitals, Bangladesh: putting infection control into contextPLOS ONE

Dear Dr. Islam,

Thank you for submitting your manuscript to PLOS ONE. After careful consideration, we feel that it has merit but does not fully meet PLOS ONE’s publication criteria as it currently stands. Therefore, we invite you to submit a revised version of the manuscript that addresses the points raised during the review process. All the comments raised by the Reviewers are relevant to improving this submission and I invited you to address them accordingly. The comments can be found at the end of this letter. Please submit your revised manuscript by 07/12/2022. If you will need more time than this to complete your revisions, please reply to this message or contact the journal office at plosone@plos.org. Please include the following items when submitting your revised manuscript:A rebuttal letter that responds to each point raised by the academic editor and reviewer(s). You should upload this letter as a separate file labeled 'Response to Reviewers'.A marked-up copy of your manuscript that highlights changes made to the original version. You should upload this as a separate file labeled 'Revised Manuscript with Track Changes'.An unmarked version of your revised paper without tracked changes. You should upload this as a separate file labeled 'Manuscript'.

We look forward to receiving your revised manuscript.

Kind regards,

Desmond Kuupiel, PhD

Academic Editor

PLOS ONE

Journal Requirements:

“We would like to thank all the study participants for their time and respect. This research was funded by the United States Centers for Disease Control and Prevention (CDC), through the cooperative agreement grant number 5U01GH1207. icddr,b acknowledges with gratitude the commitment of CDC to its research efforts. icddr,b is also grateful to the Governments of Bangladesh, Canada, Sweden, and the UK for providing core/unrestricted support. We also acknowledge Arkray Healthcare Pvt Ltd for their support.”

“This research was funded by the United States Centers for Disease Control and Prevention (CDC), through the cooperative agreement grant number 5U01GH1207. Md Saiful Islam and Emily S Gurley has received the funding.

The funders had no role in study design, data collection and analysis, decision to publish, or preparation of the manuscript. The findings and conclusions in this report are those of the authors and do not necessarily represent the official position of CDC.”

4. Please ensure that you refer to Figure 2 in your text as, if accepted, production will need this reference to link the reader to the figure

Reviewers' comments:

Reviewer's Responses to Questions

**Comments to the Author**

1. Is the manuscript technically sound, and do the data support the conclusions?

Reviewer #1: Partly

Reviewer #2: Partly

2. Has the statistical analysis been performed appropriately and rigorously? 

Reviewer #1: No

Reviewer #2: No

3. Have the authors made all data underlying the findings in their manuscript fully available?

Reviewer #1: No

Reviewer #2: Yes

4. Is the manuscript presented in an intelligible fashion and written in standard English?

Reviewer #1: Yes

Reviewer #2: Yes

5. Review Comments to the Author

Reviewer #1: General comments

The authors present the results of a well-designed study of baseline prevalence of positive TST among HCW working at the 4 largest hospitals with pulmonary units for TB patients and a prospective follow-up for those with negative baseline tests. In addition, IGRA testing was also done at one hospital using QFT Gold In-tube assay (QFT). The comparison of TST and QFT results is largely limited to kappa coefficient and ROC curve.

Specific comments

Methods –

How was the 2-step testing done? There is no mention of how many or what proportion of the individuals were positive only on the 2nd test. Were all the initial test results recorded or only those with at least 10 mm of induration? The frequency of “boosting” would be of interest in this setting.

Item that must be addressed:

There is no mention of obtaining chest radiographs on subjects with positive test results. Were these done or was only a symptom review done? The chest radiograph is considered essential to excluding subclinical TB in the US.

Results

Items that must be addressed:

Table 2 prevalence of positive TST – footnote iv and v do not indicate adjustment for age

Table 4 prevalence of positive QFT – footnotes do not include vi. No indication of adjustment for age in footnote iii.

Page 15 second sentence states the prevalence of positive TST and QFT were both associated with years of service on pulmonary wards, but this is not indicated in Table 4.

Missing analysis:

There is a consistent increase in QFT-positivity by age as one expects given the cumulative exposure in the community where they live and work. A more detailed comparison of the age specific variation in concordance and discordance of TST and QFT results by age for the 400-plus HCW with both tests would be of considerable interest. The number and proportion of results that are positive by both, positive for TST alone, QFT alone and negative for both within age categories would be of considerable interest. Those with a positive TST only, tend to be false-positive TST and those with a positive QFT alone tend to be false-positive QFT (except in older individuals) in the context of US HCW but that “tendency” based on follow-up results may not be the same for individuals within this higher-risk population. None of these immunological test results can be interpreted with full confidence of course. Presenting the data on QFT and TST in this setting should include a more detailed analysis and the authors assessment of whether IGRA are useful in this setting.

Reviewer #2: RE: PLOS1 D22-14779

“Latent TB infection in health care workers in chest diseases hospitals in Bangladesh”

This is an interesting paper describing a cross-sectional survey of health care workers in four hospitals in Bangladesh with a follow up of some (a reasonable proportion) with a repeat TST and or IGRA approximately two years later. Results are interesting in that they indicate an approximate 4% annual risk infection estimated by TST or by QFT. The authors performed two-step tuberculin testing at baseline so the TST conversion rate may be considered reasonably robust although the QFT conversion rate of 4% is certainly within the range of variability previously described for this test, particularly in high incidence settings.

Major Comments:

1. The rate of participation at one hospital was 68% at the largest hospital, but the other three smaller hospitals had a much higher participation rate. Some assessment of potential selection bias is important, for example comparing characteristics of workers who did and did not participate – initially and at the 2 year follow-up.

2. On follow up testing two years later the participation in TST was only 53% at the large hospital compared to 84% at the three smaller hospitals. Follow up for QFT was better at 77%. All of this raises again the problems of selection bias terms of who participated and who did not. At minimum, the characteristics of the workers who did or did not participate in the follow up survey should be provided, even as a supplement table. Selection bias could lead to an over or under-estimate of the conversion rate.

3. Several of the risk factors/ characteristics of workers overlap with each other. For example, type of profession is strongly correlated with level of education. Hence to show both without accounting for the other is not that meaningful. I suggest that the level of education should indicate where doctors, nurses and pharmacists lie because it is really the other workers whose level of education may be more variable that are of interest in this parameter. Similarly, age and years of work are strongly correlated. To avoid this problem of being unable to distinguish the effect of age from the effect of years of work the authors should analyze as two non-overlapping variables: (i) age of starting work at the participating hospital, and (ii) years of work at that hospital. In this way one can distinguish the effect of prior exposure until they were hired at the hospital, or exposure since starting work at the Chest hospitals that explains the high initial prevalence of infection.

4. The analysis of incident TST positivity (ie TST conversion) certain factors should not have changed over the two-year interval and should not be included in the model. For example level of education, having lived with someone with TB, years worked on pulmonary TB wards, and years of work as a HCW. What is more relevant is the exposures in the two years interval - whether they worked on the TB wards in the past two years, and if so how many hours per day. Factors that are relevant to baseline or initial prevalence but are not relevant to the incidence of infection over two years should be included in the model (ie “have they lived in the last two years with someone with TB”) which would have been a much more useful question, hence table 3 and table 5 should amended to reflect exposures specifically in the two years since they were tested and negative.

5. A major error, (as far as I can see in the data presented), is in the analysis of the initial TST reactions. Results of the first TST should be analyzed separately from results of the second TST which was to elicit the booster phenomenon. There is good evidence that the “booster phenomenon” is much less specific and much more likely affected by BCG vaccination, non-tuberculosis mycobacteria, as well as remote TB exposure. Hence analyzing initial and second TST together simply introduces misclassification into the exposure of interest which is occupational exposure. I recommend separate analyses as a required reanalysis.

6. I also recommend they drop the comparison of TST and IGRA. This is not novel. And the authors seem to suggest that the IGRA is gold standard for incident infection, when the evidence favours that after baseline two-step TST, it is the TST that is the gold standard for conversion. Obviously this is a matter of debate. Bottom line I don’t think the analysis is all that meaningful nor valuable, hence I suggest dropping table S1 and Figure 2.

Minor points:

7. I recognize that two of the authors are based at CDC and an additional one is also at a US institution. However the study is based in Bangladesh, so I believe the authors should refer to WHO recommendations for use of TST or IGRA, not US recommendations (WHO recommends they can be used interchangeably).

8. Along the same lines, perhaps also the term “latent TB infection” which is now mainly a US term should be dropped in favour of “TB infection” which WHO now uses.

6. PLOS authors have the option to publish the peer review history of their article (what does this mean?). If published, this will include your full peer review and any attached files.

Reviewer #1: No

Reviewer #2: No

---

## [Author Response · Author response to Decision Letter 0]

12 May 2023

PONE-D-22-14779

To

Desmond Kuupiel, PhD

 Academic Editor

 PLOS ONE

Dear Desmond Kuupiel,

We are thankful to the reviewers for their valuable feedback. We also thank you for allowing us to respond to the comments. Based on the helpful feedback, we revised the manuscript and believe it is more precise, clear, and informative. The following is an itemised list of our specific responses to the reviewers’ comments. We have highlighted the changes in the manuscript as well. 

We would appreciate your further review. Please contact me directly with any additional questions or comments. We look forward to hearing from you.

Sincerely

Md. Saiful Islam

Corresponding Author

Comments: Journal Requirements:

1. Please ensure that your manuscript meets PLOS ONE’s style requirements, including those for file naming. The PLOS ONE style templates can be found at

Response: Thank you. The manuscript and the file names have been updated according to PLOS ONE’s style requirements.

Comment: 2. Thank you for stating the following in the Acknowledgments Section of your manuscript:

“We would like to thank all the study participants for their time and respect. This research was funded by the United States Centers for Disease Control and Prevention (CDC), through the cooperative agreement grant number 5U01GH1207. icddr,b acknowledges with gratitude the commitment of CDC to its research efforts. icddr,b is also grateful to the Governments of Bangladesh, Canada, Sweden, and the UK for providing core/unrestricted support. We also acknowledge Arkray Healthcare Pvt Ltd for their support.”

“This research was funded by the United States Centers for Disease Control and Prevention (CDC), through the cooperative agreement grant number 5U01GH1207. Md Saiful Islam and Emily S Gurley has received the funding.

The funders had no role in study design, data collection and analysis, decision to publish, or preparation of the manuscript. The findings and conclusions in this report are those of the authors and do not necessarily represent the official position of CDC.”

Response: Thank you. We have checked the acknowledgments sections and removed any funding information from there. It should read, “We would like to thank all the study participants for their time and respect. icddr,b is grateful to the Governments of Bangladesh, Canada, Sweden, and the UK for providing core/unrestricted support. We also acknowledge Arkray Healthcare Pvt Ltd for their support.”

The funding statement is correct.

Comments: 3. Please include a separate caption for each figure in your manuscript.

Response: Separate caption for each figure has been given.

Comments: 4. Please ensure that you refer to Figure 2 in your text as, if accepted, production will need this reference to link the reader to the figure

Response: Based on the reviewer’s comments, Figure 2 has been deleted.

 Reviewers’ comments:

 Reviewer #1: General comments

 Comments: The authors present the results of a well-designed study of baseline prevalence of positive TST among HCW working at the 4 largest hospitals with pulmonary units for TB patients and a prospective follow-up for those with negative baseline tests. In addition, IGRA testing was also done at one hospital using QFT Gold In-tube assay (QFT). The comparison of TST and QFT results is largely limited to kappa coefficient and ROC curve.

Response: Thank you so much for your appreciation. Regarding the TST and QFT-GIT, a supplementary table S2 has been added, and summary findings are discussed in the manuscript as “Table S3 shows results of the analysis of concordance and discordance between the TST and the QFT-GIT test. QFT-only positive results were significantly associated with the number of years working as HCWs after controlling for sex, age of starting work, history of BCG vaccine, and history of having a household member with pulmonary TB. In contrast, TST-only positive results had no association with the exposures in the model. Both TST and QFT positive results were significantly associated with years working as HCWs after controlling for sex, age of starting work, history of BCG vaccine, and history of having a household member with pulmonary TB’, on page 14, lines 327 -334.

 Comments: Specific comments

 Methods –How was the 2-step testing done?

Response: Thank you. On page 7, lines 169-172, we added, “For person whose indurations size was < 10mm in the first-TST, we requested that person for a second test after 14 days from the first test [1]. We took their reading following the same procedure used in the first step. If the size was > 10 mm, the person was considered infected [1, 2].”

Comments: There is no mention of how many or what proportion of the individuals were positive only on the 2nd test. Were all the initial test results recorded or only those with at least 10 mm of induration? The frequency of “boosting” would be of interest in this setting.

Response: All the initial test results were recorded. On page 11, lines 251-256, we added, “Of 665 HCWs who received a two-step TST at baseline, 356 (54%) had a positive result, ranging from 39% to 67% per site. The median induration of the two-step TST was 10mm (IQR 5-13). Of the 356 two-step TST positive, 78% (277/356) were positive by a first-TST (S1). The median induration of the HCWs who were tested positive at the first-TST was 13mm (11mm-15mm) and those who were tested negative at the first-TST was 3mm (0mm-7mm). The median induration of the participants who became TST positive in the second-TST was 16.5mm (15-18)”.

 Comment: Item that must be addressed:

 There is no mention of obtaining chest radiographs on subjects with positive test results. Were these done or was only a symptom review done? The chest radiograph is considered essential to excluding subclinical TB in the US.

Response: Thank you. The primary objective of this research was to estimate the prevalence of latent TB infection among HCWs, and therefore, no chest radiographs was done on HCWs with positive TST and QFT-GIT. However, we added this as a study limitation, “Finally, we did not obtain chest radiographs on HCWs positive with TST or QFT-GIT. We only relied on TB symptoms screening and might have missed cases with active TB” On page 19, lines 449-451.

 Comments: Results

 Items that must be addressed: Table 2 prevalence of positive TST – footnote iv and v do not indicate adjustment for age.

Response: Corrected. Now it reads, “iv Adjusted for age of starting work years, history of BCG vaccine, and lived with someone diagnosed with pulmonary TB.

v Adjusted for age of starting work years, history of BCG vaccine, and lived with someone diagnosed with pulmonary TB” in Table 2.

 Comment: Table 4 prevalence of positive QFT – footnotes do not include vi. No indication of adjustment for age in footnote iii.

Response: Corrected in Table 4. We have revised it as “iii Adjusted for the history of BCG vaccine, age of starting to work in years and lived with someone diagnosed with pulmonary TB.

vi Adjusted for the age of starting to work in years and lived with someone diagnosed with pulmonary TB”.

Comment: Page 15 second sentence states the prevalence of positive TST and QFT were both associated with years of service on pulmonary wards, but this is not indicated in Table 4.

Response: The statement is true. In Table 4, HCWs working in pulmonary TB wards >20 years were 2.88 (95% CI 1.14-4.59) times higher at risk of being QFT-GIT positive when compared with HCWs working <10 years in the same wards.

 Comments: Missing analysis:

 There is a consistent increase in QFT-positivity by age as one expects given the cumulative exposure in the community where they live and work. A more detailed comparison of the age specific variation in concordance and discordance of TST and QFT results by age for the 400-plus HCW with both tests would be of considerable interest. The number and proportion of results that are positive by both, positive for TST alone, QFT alone and negative for both within age categories would be of considerable interest. Those with a positive TST only, tend to be false-positive TST and those with a positive QFT alone tend to be false-positive QFT (except in older individuals) in the context of US HCW but that “tendency” based on follow-up results may not be the same for individuals within this higher-risk population. None of these immunological test results can be interpreted with full confidence of course. Presenting the data on QFT and TST in this setting should include a more detailed analysis and the authors assessment of whether IGRA are useful in this setting.

Response: Thank you. We have added a supplementary table, S3 focusing on the concordance and discordance of TST and QFT-GIT. In the text, we said, “S3 shows results of the analysis of concordance and discordance between the TST and the QFT-GIT test. QFT only positive results were significantly associated with number of years working as HCWs after controlling for sex, age of starting to work, history of BCG vaccine, and history of having a household member with pulmonary TB, whereas TST-only positive results had no association with the exposures in the model. Having both TST and QFT positive results were significantly associated with years working as HCWs after controlling for sex, age of starting to work, history of BCG vaccine, and history of having a household member with pulmonary TB”.

Reviewer #2: RE: PLOS1 D22-14779

 Comment: “Latent TB infection in health care workers in chest diseases hospitals in Bangladesh”

 This is an interesting paper describing a cross-sectional survey of health care workers in four hospitals in Bangladesh with a follow up of some (a reasonable proportion) with a repeat TST and or IGRA approximately two years later. Results are interesting in that they indicate an approximate 4% annual risk infection estimated by TST or by QFT. The authors performed two-step tuberculin testing at baseline so the TST conversion rate may be considered reasonably robust although the QFT conversion rate of 4% is certainly within the range of variability previously described for this test, particularly in high incidence settings.

 Response: Thank you.

Major Comments:

 Comments: 1. The rate of participation at one hospital was 68% at the largest hospital, but the other three smaller hospitals had a much higher participation rate. Some assessment of potential selection bias is important, for example comparing characteristics of workers who did and did not participate – initially and at the 2-year follow-up.

Response: We added a supplementary table S2 comparing the characteristics of the participants and non-participants in the follow-up study. In the manuscript, we said, “Table S2 showed that HCWS who were lost to follow up for TST statistically significantly varied by location of hospitals, education, and age of starting to work’ on pages 12-13, lines 285-287.

On page 14, lines 316-19, we added, “Table S2 showed that there was no statistically significant difference in HCWs’ characteristics who participated in the follow-up QFT-GIT and who were lost to follow up”.

In the limitation, we added, “A second limitation was participation attrition; 35% of HCWs were lost to follow up for the repeat TST and 23% for the repeat QFT-GIT due to job transfer, illness, retirement from work, and unwillingness to participate. These participants lost to follow-up could have developed a new infection, and therefore, our study could have inaccurately estimated the true incidence of TST and QFT-GIT positivity. In addition, owing to the loss of follow-up, the incidence analysis was underpowered to detect true differences between groups, particularly by professional category” on page 19, lines 443-449.

We do not have information from HCWs who did not participate in the baseline study, and therefore, it was beyond the scope of the study to compare the characteristics between participants and non-participants at baseline.

 Comment: 2. On follow up testing two years later the participation in TST was only 53% at the large hospital compared to 84% at the three smaller hospitals. Follow up for QFT was better at 77%. All of this raises again the problems of selection bias terms of who participated and who did not. At minimum, the characteristics of the workers who did or did not participate in the follow up survey should be provided, even as a supplement table. Selection bias could lead to an over or under-estimate of the conversion rate.

Response: We added a supplementary table S2 comparing the characteristics of the participants and non-participants in the follow-up study. In the manuscript, we said, “Table S2 showed that HCWS who were lost to follow up for TST statistically significantly varied by location of hospitals, education, and age of starting to work’ on pages 12-13, lines 285-287.

On page 14, lines 317-19, we added, “Table S2 showed that there was no statistically significant difference in HCWs’ characteristics who participated in the follow-up QFT-GIT and who were lost to follow up”.

In the limitation, we added, “A second limitation was participation attrition; 35% of HCWs were lost to follow up for the repeat TST and 23% for the repeat QFT-GIT due to job transfer, illness, retirement from work, and unwillingness to participate. These participants lost to follow-up could have developed a new infection, and therefore, our study could have inaccurately estimated the true incidence of TST and QFT-GIT positivity. In addition, owing to the loss of follow-up, the incidence analysis was underpowered to detect true differences between groups, particularly by professional category” on page 19, lines 443-449.

Comment: 3. Several of the risk factors/ characteristics of workers overlap with each other. For example, type of profession is strongly correlated with level of education. Hence to show both without accounting for the other is not that meaningful. I suggest that the level of education should indicate where doctors, nurses and pharmacists lie because it is really the other workers whose level of education may be more variable that are of interest in this parameter. 

Response: We would like to clarify that we assessed the variance inflation factor (VIF) that measures how the variance of an independent variable is influenced or inflated due to its correlation or interaction with other independent variables in the models, and risk factors with VIF of more than 5 were examined for collinearity and dropped from the models [3]. We did not find any collinearity between education and the category of a profession.

Comment: Similarly, age and years of work are strongly correlated. To avoid this problem of being unable to distinguish the effect of age from the effect of years of work the authors should analyse as two non-overlapping variables: (i) age of starting work at the participating hospital, and (ii) years of work at that hospital. In this way one can distinguish the effect of prior exposure until they were hired at the hospital, or exposure since starting work at the Chest hospitals that explains the high initial prevalence of infection.

Response: Thank you. Based on your suggestion, we have now used the variable “Age of starting work” instead of age and “years of work in the hospital”. 

 Comments: 4. The analysis of incident TST positivity (ie TST conversion) certain factors should not have changed over the two-year interval and should not be included in the model. For example, level of education, having lived with someone with TB, years worked on pulmonary TB wards, and years of work as a HCW. What is more relevant is the exposures in the two years interval - whether they worked on the TB wards in the past two years, and if so how many hours per day. Factors that are relevant to baseline or initial prevalence but are not relevant to the incidence of infection over two years should be included in the model (ie “have they lived in the last two years with someone with TB”) which would have been a much more useful question, hence table 3 and table 5 should amended to reflect exposures specifically in the two years since they were tested and negative.

Response: Thank you. We agree with the reviewer that the two years of exposure could have been more helpful. However, we did not collect the two years exposures separately and therefore have not been used in the model. 

 Comment: 5. A major error (as far as I can see in the data presented), is in the analysis of the initial TST reactions. Results of the first TST should be analysed separately from results of the second TST which was to elicit the booster phenomenon. There is good evidence that the “booster phenomenon” is much less specific and much more likely affected by BCG vaccination, non-tuberculosis mycobacteria, as well as remote TB exposure. Hence analysing initial and second TST together simply introduces misclassification into the exposure of interest which is occupational exposure. I recommend separate analyses as a required reanalysis.

Response: Thank you for your concern. We now analysed the results of the first TST separately and added the results as a supplementary table S1.

We want to clarify that the TST analysis and the interpretation have been done following the guidance given by Nayak and Acharjya (2012)[4], and this statement is now added on page 8, lines 187-189. The article Nayak and Acharjya (2012) says regarding booster effect as, “In some persons who are infected with M. tuberculosis, the ability to react to tuberculin may wane over time. When given TST years after infection, these persons may have a false-negative reaction. However, the TST may stimulate the immune system, causing a positive or boosted reaction to subsequent tests. Giving a second TST after an initial negative TST reaction is called two-step testing. When sensitisation to mycobacteria has occurred many years earlier, an initial intradermal injection of tuberculin may produce a negative or weakly positive response due to there being too few sensitised lymphocytes in circulation to produce a significant local response. If the test is repeated, a larger reading may be obtained due to the immune response being ‘recalled ‘or ‘boosted’ by the first test. The second boosted reading is the correct one – that is, the result that should be used for decision-making or future comparison. Boosting is maximal if the second test is placed between one and five weeks after the initial test, and it may continue to be observed for up to two years.”

Moreover, two-step TST is recommended to avoid misclassification of subsequent positive TSTs as a TST conversion, indicating recent infection when they are actually a result of boosting[5].

 6. I also recommend they drop the comparison of TST and IGRA. This is not novel. And the authors seem to suggest that the IGRA is gold standard for incident infection, when the evidence favours that after baseline two-step TST, it is the TST that is the gold standard for conversion. Obviously this is a matter of debate. Bottom line I don’t think the analysis is all that meaningful nor valuable, hence I suggest dropping table S1 and Figure 2.

Response: Thank you. Based on your suggestion, we have dropped Table S1 and Figure 2. 

 Minor points:

 Comments: 7. I recognise that two of the authors are based at CDC and an additional one is also at a US institution. However, the study is based in Bangladesh, so I believe the authors should refer to WHO recommendations for use of TST or IGRA, not US recommendations (WHO recommends they can be used interchangeably).

Response: We have now deleted this from page 5, lines 113-119, “The US Centers for Disease Control and Prevention (CDC) and the Infectious Diseases Society of America/American and Thoracic Society (ATS) clinical guidelines recommend using an IGRA such as QuantiFERON®-TB Gold In-Tube test (QFT-GIT) rather than TST to test adults who are likely to be infected with M. tuberculosis, have a low or medium risk of disease progression, or have a history of BCG vaccination [6]. Guidelines recommend TST when the laboratory capacity required for an IGRA is unavailable [6].”

 Comments: 8. Along the same lines, perhaps also the term “latent TB infection” which is now mainly a US term should be dropped in favour of “TB infection” which WHO now uses.

Response: Many WHO guidelines still use latent TB infection in their policies and guidelines[7]. Therefore, we kept the term “latent TB infection”.

References:

1. Lien, L.T., et al., Prevalence and risk factors for tuberculosis infection among hospital workers in Hanoi, Viet Nam. PLoS One, 2009. 4(8): p. e6798.

2. Anonymous, Procedure for administering, reading and interpreting Mantoux tuberculin skin tests to detect infection with M. tuberculosis 2001.

3. Kwon, Y.S., et al., Factors that Predict Negative Results of QuantiFERON-TB Gold In-Tube Test in Patients with Culture-Confirmed Tuberculosis: A Multicenter Retrospective Cohort Study. PLoS One, 2015. 10(6): p. e0129792.

4. Nayak, S. and B. Acharjya, Mantoux test and its interpretation. Indian Dermatol Online J, 2012. 3(1): p. 2-6.

5. Lewinsohn, D.M., et al., Official American Thoracic Society/Infectious Diseases Society of America/Centers for Disease Control and Prevention Clinical Practice Guidelines: Diagnosis of Tuberculosis in Adults and Children. Clinical Infectious Diseases, 2016. 64(2): p. e1-e33.

6. Lewinsohn, D.M., et al., Official American Thoracic Society/Infectious Diseases Society of America/Centers for Disease Control and Prevention Clinical Practice Guidelines: Diagnosis of Tuberculosis in Adults and Children. Clin Infect Dis, 2017. 64(2): p. 111-115.

7. World Health Organization, WHO consolidated guidelines on tuberculosis Module 1: Prevention-tuberculosis preventative treatment. 2020: Geneva, Switzerland.

---

## [Decision Letter · Decision Letter 1]

17 Jul 2023

PONE-D-22-14779R1Prevalence and incidence of latent tuberculosis infection among healthcare workers in chest diseases hospitals, Bangladesh: putting infection control into contextPLOS ONE

Dear Dr. Islam,

Thank you for submitting your manuscript to PLOS ONE. After careful consideration, we feel that it has merit but does not fully meet PLOS ONE’s publication criteria as it currently stands. Therefore, we invite you to submit a revised version of the manuscript that addresses the points raised during the review process.

We look forward to receiving your revised manuscript.

Kind regards,

Frederick Quinn

Academic Editor

PLOS ONE

Journal Requirements:

Reviewers' comments:

Reviewer's Responses to Questions

**Comments to the Author**

1. If the authors have adequately addressed your comments raised in a previous round of review and you feel that this manuscript is now acceptable for publication, you may indicate that here to bypass the “Comments to the Author” section, enter your conflict of interest statement in the “Confidential to Editor” section, and submit your "Accept" recommendation.

Reviewer #1: All comments have been addressed

Reviewer #2: (No Response)

2. Is the manuscript technically sound, and do the data support the conclusions?

Reviewer #1: Yes

Reviewer #2: Partly

3. Has the statistical analysis been performed appropriately and rigorously? 

Reviewer #1: Yes

Reviewer #2: Yes

4. Have the authors made all data underlying the findings in their manuscript fully available?

Reviewer #1: No

Reviewer #2: Yes

5. Is the manuscript presented in an intelligible fashion and written in standard English?

Reviewer #1: Yes

Reviewer #2: Yes

6. Review Comments to the Author

Reviewer #1: (No Response)

Reviewer #2: Comment to the authors re P122-14779R1

I initially reviewed this paper some months ago. I found it had many strengths, particularly the careful performance of two-step tuberculin testing and QuantiFERON testing at baseline, and follow-up among employees at one of the four participating hospitals. Plus the use of questionnaire information to look at risk factors for baseline prevalent positive tests and incident positive tests. By and large, the authors have made most important revisions but there remain two issues that must be addressed.

1. Two-step tuberculin testing

I disagree with the authors in their continued assertion that it is the final result of the two-step test that is the most precise and valuable from an epidemiologic point of view. They cite a paper published in the Indian Journal of Dermatology which, with all due respect to the two authors of that review, provides incorrect information. There is good evidence that boosting (or a positive second TST shortly after a negative first TST) is related to much more than remote TB exposure but also to BCG vaccination and non-tuberculosis mycobacterial exposure, both of which would have been common in this study population. There was also good evidence that the risk of disease is lower in those with a positive second test compared to the first, which doubtless relates to this lack of specificity. They have performed the analysis I suggested, namely factors associated with the first tuberculin test positivity but relegate this to the supplement. I suggest that the primary analysis should be factors associated with a positive first TST, and the factors associated with the positive second test (ie the booster phenomenon) are in the supplement. The analysis of factors associated with any positive test at baseline should be dropped completely. I personally consider this mandatory. Perhaps the co-authors from CDC can suggest other reviews and guidance on the topic of two-step TST and interpretation and management.

2. Past TB disease exclusion

A somewhat more minor point but I think also should be considered strongly by the authors is that the persons with past TB disease were excluded from analyses. This does not really make sense when they are interested in occupational tuberculosis infection. TB disease is the most important manifestation of occupational TB exposure and so these individuals should have been included. If tuberculin testing or QuantiFERON was not done, I would simply assume that they would have had a positive test and classify them as having TB infection. This is only 2% of participants but nevertheless this is an important group and should not be excluded.

Minor points:

1. Exposure in hospitals is not just related to the TB ward or to patients with recognized or diagnosed TB. Of course, in Chest Hospitals all patients who are admitted may be screened for TB disease and so the likelihood that patients on other “non-TB” wards actually have active TB disease that is unrecognized may be lower. However in most general hospitals most TB exposures occur from patients who are admitted for other reasons such as “pneumonia”, have unrecognized TB disease. These individuals are not treated, nor isolated and so represent the greatest hazard to workers. Perhaps the authors can comment on this possibility, had they looked at how many patients with TB disease are in fact admitted to other services or other wards.

2. Latent TB Infection VS TB Infection – The authors prefer to stick with LTBI, given that several co-authors are from CDC;they suggest that WHO documents still refer to LTBI – this is incorrect. Older WHO documents may use the term LTBI but current WHO terminology is TB Infection.

3. Burden VS Incidence – “TB burden” reflects population size and incidence. Hence, countries with very large population size but intermediate incidence may have high burden (e.g., China) but for occupational exposure and risk of infection, I think risk is driven by incidence rather than burden.

4. The authors incorrectly state that IGRAS are not affected by immunosuppression, in fact sensitivity is reduced by immunosuppression, particularly HIV infection, particularly for QuantiFERON, which is the test they have used here. In systematic reviews there is little difference in the sensitivity of QuantiFERON VS TST in immunocompromised populations.

5. They also state that IGRAs are less affected by NTM. There is theoretical evidence that NTM may affect IGRAS less but in reality, there is no actual epidemiological data that I am aware of that shows this. Can the authors provide a citation for a study showing this?

7. PLOS authors have the option to publish the peer review history of their article (what does this mean?). If published, this will include your full peer review and any attached files.

Reviewer #1: No

Reviewer #2: **Yes: **Richard Menzies

---

## [Author Response · Author response to Decision Letter 1]

4 Aug 2023

PONE-D-22-14779R1

To

Frederick Quinn

Academic Editor

 PLOS ONE

Dear Frederick Quinn,

We are thankful to the reviewers for their 2nd round of comments and feedback. We also thank you for allowing us to respond to the comments. Based on the helpful feedback, we revised the manuscript and believe it is more precise, clear, and informative. The following is an itemised list of our specific responses to the reviewers’ comments. We have highlighted the changes in the manuscript as well. 

We would appreciate your further review. Please contact me directly with any additional questions or comments. We look forward to hearing from you.

Sincerely

Md. Saiful Islam

Corresponding Author

Comment: Reviewer #1: (No Response)

Response: Thank you.

Comment: Reviewer #2: Comment to the authors re P122-14779R1

I initially reviewed this paper some months ago. I found it had many strengths, particularly the careful performance of two-step tuberculin testing and QuantiFERON testing at baseline, and follow-up among employees at one of the four participating hospitals. Plus the use of questionnaire information to look at risk factors for baseline prevalent positive tests and incident positive tests. By and large, the authors have made most important revisions but there remain two issues that must be addressed.

Response: Thank you for your appreciation.

Comment: 1. Two-step tuberculin testing

I disagree with the authors in their continued assertion that it is the final result of the two-step test that is the most precise and valuable from an epidemiologic point of view. They cite a paper published in the Indian Journal of Dermatology which, with all due respect to the two authors of that review, provides incorrect information. There is good evidence that boosting (or a positive second TST shortly after a negative first TST) is related to much more than remote TB exposure but also to BCG vaccination and non-tuberculosis mycobacterial exposure, both of which would have been common in this study population. There was also good evidence that the risk of disease is lower in those with a positive second test compared to the first, which doubtless relates to this lack of specificity. They have performed the analysis I suggested, namely factors associated with the first tuberculin test positivity but relegate this to the supplement. I suggest that the primary analysis should be factors associated with a positive first TST, and the factors associated with the positive second test (ie the booster phenomenon) are in the supplement. The analysis of factors associated with any positive test at baseline should be dropped completely. I personally consider this mandatory. Perhaps the co-authors from CDC can suggest other reviews and guidance on the topic of two-step TST and interpretation and management.

Response: Thank you so much for your valuable input. We have now revised the analysis and used one-step TST as primary analysis. Based on the revised analysis, we have updated the results section. On pages 25-26, lines 595-610, we updated the Table 2:Factors associated with one-step-TST positivity among HCWs in four chest diseases hospitals, Bangladesh, 2013 

 On pages 10-11, lines 231-259, we also updated the results section as-

“Prevalence of TST Positivity

 Of 731 HCWs who received a one-step TST at baseline, 296 (40%) had a positive result, ranging from 23% to 46% per site. The median induration of the one-step TST was 8mm (IQR 12). Of the 435 HCWs who were negative at one-step TST, 392 (90%) received two-step TST of whom 21% (81/392) were positive (Table S1). The median induration of the HCWs who were tested positive at the one-step TST was 13mm (IQR10mm-15mm) and those who were tested negative at the first-TST was 3mm (IQR 0mm-7mm). The median induration of the participants who became TST positive in the second-TST was 12mm (IQR11-13). The one-step TST prevalence among HCWs working in Hospital C was 46% (31/68), significantly higher (adjusted odds ratio [aOR]=3.58, 95% CI: 1.61–7.96) than the other hospitals (Table 2). 

In bivariate analyses, study sites, years worked on pulmonary TB wards, longer duration of stay in pulmonary TB wards per day, and longer duration of service as a HCW were significantly associated with two-step TST positivity at baseline. Age of starting work, sex, previous history of BCG vaccination, and ever having had a household member with PTB were not associated with TST positivity (Table 2). In the multivariable models, the HCWs at hospital C (aOR=3.58, 95% CI: 1.61–7.96), hospital A (aOR=3.08, 95% CI: 1.37–6.90), and hospital D (aOR=3.23, 95% CI: 1.70–6.12) had higher odds of positive TST results compared with HCWs at Hospital B (Table 2). Also, HCWs with a history of working 11–20 years and >20 years on pulmonary TB patients wards had significantly higher odds of TST positivity (aOR=2.01, 95% CI: 1.39–2.93 and aOR=2.09, 95% CI: 1.26–3.45) compared with HCWs working less than 10 years on the same wards (Table 2). Moreover, HCWs spending more than two hours a day in pulmonary TB wards had significantly higher odds of TST positivity (aOR=1.91, 95% CI: 1.37-2.68). Duration of service as a HCW 11-20 years (aOR 2.09, 95% CI: 1.48-2.97) and >20 years (aOR 2.05, 95% CI: 1.28-3.28) were also associated with higher odds of TST positivity compared with those ≤10 years(Table 2)”.

We have analysed the two-step TST results separately and added as a supplementary table1 on page 31. 

On page 7 line 164, we also deleted the sentence, “TST analysis and the interpretation have been done following the guidance given by Nayak and Acharjya (2012)” and removed the citation form the Indian Journal of Dermatology. 

Comment: 2. Past TB disease exclusion

A somewhat more minor point but I think also should be considered strongly by the authors is that the persons with past TB disease were excluded from analyses. This does not really make sense when they are interested in occupational tuberculosis infection. TB disease is the most important manifestation of occupational TB exposure and so these individuals should have been included. If tuberculin testing or QuantiFERON was not done, I would simply assume that they would have had a positive test and classify them as having TB infection. This is only 2% of participants but nevertheless this is an important group and should not be excluded.

Response: Thank you. The HCWs with a history of past TB diseases has been included in the analysis and the tables and the manuscript have been updated based on the revised analysis.

Comment: Minor points: 1. Exposure in hospitals is not just related to the TB ward or to patients with recognized or diagnosed TB. Of course, in Chest Hospitals all patients who are admitted may be screened for TB disease and so the likelihood that patients on other “non-TB” wards actually have active TB disease that is unrecognized may be lower. However, in most general hospitals most TB exposures occur from patients who are admitted for other reasons such as “pneumonia”, have unrecognized TB disease. These individuals are not treated, nor isolated and so represent the greatest hazard to workers. Perhaps the authors can comment on this possibility, had they looked at how many patients with TB disease are in fact admitted to other services or other wards.

Response: We did not have data on how many patients with TB disease were admitted to what services or wards and therefore it is beyond the scope of the study.

Comment: 2. Latent TB Infection VS TB Infection – The authors prefer to stick with LTBI, given that several co-authors are from CDC;they suggest that WHO documents still refer to LTBI – this is incorrect. Older WHO documents may use the term LTBI but current WHO terminology is TB Infection.

Response: Thank you. LTBI has been changed to TB infection (TBI) throughout the manuscript.

Comments: 3. Burden VS Incidence – “TB burden” reflects population size and incidence. Hence, countries with very large population size but intermediate incidence may have high burden (e.g., China) but for occupational exposure and risk of infection, I think risk is driven by incidence rather than burden.

Response: We have deleted the word burden and revised the sentence. Now it reads, “In collaboration with NTP, we sought to estimate the prevalence and incidence of TBI using both IGRA and TST and associated risk factors among HCWs in Bangladeshi chest disease hospitals and identify sub-populations of HCWs at the highest risk of TBI who can be targeted by TB IPC interventions.” On page 5 lines 113-121.

Comment: 4. The authors incorrectly state that IGRAS are not affected by immunosuppression, in fact sensitivity is reduced by immunosuppression, particularly HIV infection, particularly for QuantiFERON, which is the test they have used here. In systematic reviews there is little difference in the sensitivity of QuantiFERON VS TST in immunocompromised populations.

Response: Thank you. We have revised the sentence. Now, it reads, “IGRAs are performed in vitro and have a number of advantages over TST, including no requirement of return visits and being less affected by exposures to BCG vaccination, or by NTM infection” on page 5 lines 105-107.

Comment: 5. They also state that IGRAs are less affected by NTM. There is theoretical evidence that NTM may affect IGRAS less but in reality, there is no actual epidemiological data that I am aware of that shows this. Can the authors provide a citation for a study showing this?

Response: Thank you. We have now added this reference to support our statement, “Hermansen, T.S., et al., Non-tuberculous mycobacteria and the performance of interferon gamma release assays in Denmark. PLoS One, 2014. 9(4): p. e93986 on page 5 in 107”

---

## [Decision Letter · Decision Letter 2]

31 Aug 2023

Prevalence and incidence of tuberculosis infection among healthcare workers in chest diseases hospitals, Bangladesh: putting infection control into context

PONE-D-22-14779R2

Dear Dr. Islam,

We’re pleased to inform you that your manuscript has been judged scientifically suitable for publication and will be formally accepted for publication once it meets all outstanding technical requirements.

Kind regards,

Frederick Quinn

Academic Editor

PLOS ONE

Additional Editor Comments (optional):

Reviewers' comments:

Reviewer's Responses to Questions

**Comments to the Author**

1. If the authors have adequately addressed your comments raised in a previous round of review and you feel that this manuscript is now acceptable for publication, you may indicate that here to bypass the “Comments to the Author” section, enter your conflict of interest statement in the “Confidential to Editor” section, and submit your "Accept" recommendation.

Reviewer #2: All comments have been addressed

2. Is the manuscript technically sound, and do the data support the conclusions?

Reviewer #2: (No Response)

3. Has the statistical analysis been performed appropriately and rigorously? 

Reviewer #2: (No Response)

4. Have the authors made all data underlying the findings in their manuscript fully available?

Reviewer #2: (No Response)

5. Is the manuscript presented in an intelligible fashion and written in standard English?

Reviewer #2: (No Response)

6. Review Comments to the Author

Reviewer #2: (No Response)

7. PLOS authors have the option to publish the peer review history of their article (what does this mean?). If published, this will include your full peer review and any attached files.

Reviewer #2: **Yes: **Dick Menzies

---

## [Editor Report · Acceptance letter]

18 Sep 2023

PONE-D-22-14779R2 

Prevalence and incidence of tuberculosis infection among healthcare workers in chest diseases hospitals, Bangladesh: putting infection control into context 

Dear Dr. Islam:

I'm pleased to inform you that your manuscript has been deemed suitable for publication in PLOS ONE. Congratulations! Your manuscript is now with our production department. 

Kind regards, 

on behalf of

Dr. Frederick Quinn 

Academic Editor

PLOS ONE